# Can Blockchain Be a Basis to Ensure Transparency in an Agricultural Supply Chain?

Sarah Katharina Kraft and Florian Kellner *

Faculty of Business, Economics and Management Information Systems, University of Regensburg,
93053 Regensburg, Germany; sarah-katharina.kraft@ur.de
* Correspondence: florian.kellner@ur.de

**Abstract:** Many supply chains within developing countries lack transparency and are fraught with fraud, corruption, and a substantial number of intermediaries. For several decades, the cocoa sector has faced multiple social, economic, and environmental challenges, some of which include the issue of child labor and very low incomes for farmers, leading to poor living conditions. Blockchain technology has a high potential to reduce—or completely eradicate—some of these hurdles. In this article, we present a blockchain-based solution based on the open-source framework Hyperledger Fabric for the cocoa supply chain to promote transparency and reduce fraud. In doing so, we explicitly describe how farmers can be directly integrated into the whole blockchain solution considering the limited infrastructure, knowledge, and technologies available to them. Since about 70% of all cocoa worldwide is produced in West Africa, this case study uses the cocoa sector in Ghana as an example.

**Keywords:** sustainability; cocoa; blockchain; supply chain; transparency

## 1. Introduction

Chocolate, which derives its flavor from cocoa beans, is a popular luxury food worldwide and enjoyed in different varieties. Cocoa beans are among the ten most-exported commodities in the world and are imported by industrialized nations. The Netherlands is in the first place on the list of the largest importers, followed by Germany and the USA [1,2]. In turn, Germany exports the most finished chocolate products. To meet these production needs, Germany imports 90% of its cocoa from West Africa, where over 70% of the world's cocoa beans are grown and harvested by Côte d'Ivoire and Ghana [2,3].

The poor living and working conditions of the cocoa farmers, who form the basis of the cocoa supply chain, have been the focus of criticism in recent times [4,5]. Voluntary international initiatives have emerged to bring an end to human rights abuses in the supply chain networks, such as child labor or unfair wages. These efforts include, for example, the Harkin–Engel Protocol, in which companies promised to eliminate severe child labor by the year 2005, or the United Nations (UN) Guiding Principles on Business and Human Rights published in 2011, as well as the published Guiding Principles of the Organisation for Economic Cooperation and Development (OECD) [6–8]. Despite these and other voluntary actions, there has been no significant improvement in the living and working conditions of cocoa farmers and their families over the past 15 to 20 years [9].

The cocoa supply chain is complex with a lack of transparency. Thus, buying companies rarely know the origin of the cocoa beans. For instance, it is usually unknown whether the cocoa beans come from illegal plantations or whether children were engaged in the harvesting process [10]. Apart from the poor living standards of the cocoa farmers caused by low and volatile cocoa prices, the increasing demand for cocoa from Western countries also leads to environmental issues such as deforestation due to the urge to grow more cocoa [5,10,11].

The social and environmental impacts of the production process such as child labor, corruption, safety, and the health conditions of cocoa farmers have become increasingly

important to consumers in recent years as they demand more insight into the supply chain and related processes of products. Since the agri-food supply chain is mainly burdened by intermediaries, it lacks transparency, leaving consumers with little information about the food's origin and the production process [5,12–14].

Transparent supply chains that offer more information on the origin of products allow companies to connect well with consumers, build trust, and have an improved overview of each component of the supply chain. To achieve this, technologies need to be developed and implemented to promote transparency [14,15]. Transparency, in this vein, means a level of common understanding and access to information about the product, such as labor circumstances, transactions, production process, or environmental impacts, which are requested from actors involved in an agri-food supply chain. The information requested needs to be complete, accurate, and without any distortion and irrelevant details. Hence, the prerequisite for a transparent agri-food supply chain is to know what information actors need [16,17].

Existing tracking technologies such as radio frequency identification (RFID) tags, sensors, or barcodes that collect and store data along supply chains are limited, for instance, by centralized data or non-immutable data/transactions, especially in environments that are less trustworthy. A blockchain can provide a fully auditable and valid ledger of transactions, and through its encryption and control mechanism, transparency is safeguarded by storing information in such a way that it cannot be altered without recording the changes [14,15,18].

Many of the current challenges in the agri-food supply chain that affect the farmers could be tackled right from the origin. Nevertheless, none of the existing solutions directly involve smallholder farmers during their design and implementation, for instance, in West Africa, where almost all of the raw products are produced at a small scale. This work is original in the sense that it offers a solution to the prevailing challenges by which farmers are directly incorporated.

In this context, our paper aims to answer the following research question: Does it make sense to implement blockchain technology to promote transparency in the cocoa supply chain?

To address this question, (1) we investigated blockchain technology in agri-food supply chains and present an exemplary case study from the Ghanaian cocoa sector. Our results are based on expert interviews conducted with different Ghanaian actors along the cocoa supply chain. (2) We developed a blockchain-based solution for the cocoa supply chain based on the Hyperledger Fabric framework to promote the transparency and immutability of transactions. The uniqueness of our approach is that it directly involves the cocoa farmers, considering their limited infrastructure, knowledge, and technologies. Our research is to establish a basis to help fight corruption and bring about fairness in the cocoa sector, especially in the country of origin.

Our paper is organized as follows: Section 2 presents the potential of blockchain technology in supply chains and demonstrates the implementation of blockchain applications in the agri-food sector. Section 3 introduces the cocoa supply chain and the situation of the cocoa farmers in Ghana. Based on this, we show our blockchain-based solution for the cocoa supply chain. In Section 4, we discuss the presented solution. Section 5 contains concluding remarks.

## 2. Literature Review

### 2.1. About the Potential of Blockchain Technology in Supply Chains

In 2008, blockchain technology became known with the emergence of the cryptocurrency Bitcoin [19,20]. Blockchain technology does not only have a high potential in finance, but also a high ability to overcome challenges associated with supply chains such as lack of trust, risk of fraud, and counterfeiting. Moreover, the lack of transparency of all transactions across a supply chain and the authenticity of a product can be curbed using blockchain technology [21,22].

The three most significant characteristics of blockchain technology, especially for supply chains, are immutability, decentralization, and audibility [23–25]. As of today, it is technologically impossible to change or manipulate data stored in a blockchain, such as executed transactions. If a transaction is subsequently changed, the hash value of the block concerned would also change. To conceal the manipulation, all subsequent blocks and their hash values would have to be adjusted, including the copies at all nodes/participants, for which a cryptographic puzzle would have to be solved. For this reason, the blockchain is tamper-proof or immutable. In this context, tamper-proof means that once the data, for example, a transaction, are stored in the blockchain, they cannot be deleted, altered, or falsified. This tamper-proofing of data can improve transparency in supply chains by giving participants in the blockchain network-controlled access to these data [24,26,27].

In a blockchain network, there is no need for a central authority to execute transactions, as the nodes decide on the documentation of transactions inside the chain. Processes can be automated using smart contracts in the blockchain network, as smart contracts apply and execute business logic. Thus, it eliminates the need for third parties, such as intermediaries, within a supply chain to execute transactions. Decentralized data storage reduces the risk of failure, unlike centralized storage with a single point of access [23,24,28,29]. Each transaction in a blockchain is characterized by a timestamp and other information, such as a key. This type of record and visibility for all participants in the blockchain network allows for easy audibility and traceability of transactions. This feature is particularly beneficial in supply chains, allowing, for instance, the origin and the manufacturing process of products or food to be traced in real-time [15,23,25,30].

Because of the blockchain's characteristics, it is very attractive to apply it in supply chains to gather information about each product's transaction and the different actors in real-time through the entire value chain. Since every actor in a blockchain has access to all the information stored in this network, transparency across the whole supply chain will be promoted. This leads to less fraud and corruption. Moreover, the interaction among different actors within the supply chain can be improved [22,31].

*2.2. Blockchain Applications in Agri-Food Supply Chains*

Supply chains in the agri-food sector show high complexity as there is typically a great number of stakeholders, which leads to difficulty in approving important data such as the country of origin, safety, and quality issues across the whole supply chain [12,32]. Transparency and trustworthiness in the supply chain of food products are an important undertaking due to food safety and contamination risks (think of, for example, the milk scandal in China in 2008 or the horse meat scandal in Europe in 2013) and rising environmental and social demands placed on palm oil and other food products, for instance [33–36]. Blockchain technology has a proven track-record to ensure the transparency, traceability, and immutability of transactions for agri-food supply chains to promote food safety or provide (transparent) information, for example, about the product's origin across the whole supply chain [13,32].

In 2018, Walmart and Carrefour started their pilot projects to use blockchain technology to trace and track agricultural products such as mango and pork from China [37,38]. Since 2019, Walmart has succeeded in using blockchain technology for over 25 food products, whereby all suppliers of leafy greens are obliged to use blockchain technology for tracing. Based on the blockchain, Walmart has benefited from having to trace its food and provide a transparent process all along the supply chain of its food products, making it possible to detect, for instance, the origin of a food-borne disease faster should there be an outbreak. Moreover, Walmart promotes food safety through a higher transparency process, making the food certificates more trustworthy [37,39]. Till today, Carrefour uses blockchain technology for over seven food products and is planning to extend it to cover all food products coming from the label Auvergene Filière Qualitè Carrefour by 2022. Carrefour's interest in using the blockchain is to assure their customers about the safety of their food as achieved via traceability and transparency. In doing so, it can fully inform its customers

about the product's journey, starting with the food's origin [40]. Both cases show that blockchain technology can be beneficial in agri-food supply chains. Besides Walmart and Carrefour, there are many other use cases of blockchain in agri-foods, such as tuna from New Zealand or wheat from China (cf. Table 1).

Furthermore, the study of Bux et al. [41] investigated the opportunities and limitations of blockchain technology in the supply chain of halal food. To improve the safety and security of halal food, the authors discussed constraints like standards and opportunities such as improving trust and traceability, as well as creating standard procedures and secure hygiene. Besides this, Bux et al. [41] demonstrated how blockchain technology can also contribute to social and environmental sustainability, for example, by reducing chemical product usage in agriculture and manufacturing or increasing the level of customer trust [41].

**Table 1.** Examples of blockchain use cases in agri-food supply chains.

| Agri-Food | Country of Origin | References |
|-----------|-------------------|------------|
| Coffee | Colombia, Ethiopia, Honduras | [42–44] |
| Egg | USA | [45] |
| Soybean | USA | [32,46] |
| Tuna | New Zealand | [47] |
| Turkey | USA | [48] |
| Wheat | China | [49] |

Blockchain technology can be implemented based on open-source frameworks such as *Ethereum* or *Hyperledger Fabric* or using dedicated software solutions such as *IBM Food Trust*, *farmer connect*, *Agriledger*, or *Grain Chain*, which are particularly designed for agri-food supply chains [43,50–54]. Nevertheless, an outstanding barrier to applying blockchain technology to real-life solutions is the implementation in supply chain networks with low technological infrastructure [55].

Consequently, one drawback of most software solutions is that they are either designed for or used in developed countries, even though the raw materials are mostly found in developing countries. This means that most software solutions are trenched by their inability to function in developing countries because of the unavailability of devices or infrastructures (such as internet connections that barely exist or are very slow due to poorly developed telecommunication networks) to effectively function. Another fact is that peasant farmers who directly produce the raw materials are left out or are poorly integrated because of several factors such as low education and inaccessibility. Furthermore, most software solutions are too complex and expensive for (smallholder) farmers [14,56,57].

### 2.3. Blockchain Applications in the Cocoa Supply Chain

Arsyad et al. [58,59] and Chong et al. [60] showed how blockchain technology can be implemented in the cocoa supply chain. Arsyad et al. [58] suggested applying a two-factor blockchain based on digital watermarking techniques to record and validate the data of cocoa production using images. The authors showed a simulation scenario with no practical data inputs from cocoa production [58]. Chong et al. [60] developed a blockchain-based solution for cocoa export supply chains in Peru with the purpose of reducing the hurdle of obtaining real-time information from the international markets for Peruvian farmers. The authors used *Hyperledger Composer*, which has been outdated since 2019 [61]. Arsyad et al. [59] proposed via a farm simulation an *Encapsulating Block Mesh* based on blockchain technology and *the bucket principle* implemented by a Modular Block Chain sensing instrument to ensure transparent data documentation in the cocoa supply chain, especially for farm activities. Their suggestion was based on an abstract open simulator game, without inputs and consideration of the real cocoa production and its environment or supply chain [59].

In addition, neither Arsyad et al. [58,59] nor Chong et al. [60] described how exactly the farmers can be integrated into the blockchain, even though the farmers are the basis of the whole supply chain. Moreover, the prior research projects did not point out possible challenges and recommendations for implementation. Lastly, even though the prior solutions were suggested for cocoa supply chains, they cannot be implemented everywhere and, for example, in West Africa, where over 70% of the world's cocoa is produced, since the environmental and political situations are different [3,58,60].

The integration of the cocoa farmer into blockchain technology is very important because, without this, complete transparency in the supply chain cannot be achieved. Further, fraud and corruption by middlemen cannot be reduced and fair payments for the farmers cannot be guaranteed.

## 3. Blockchain-Enabled Traceability in the Cocoa Supply Chain: A Case Study of Cocoa Farmers in Ghana

In 2020, more than three million tons of cocoa beans were exported worldwide, out of which 90% originated from approximately six million small-scale farms. Farmers from two West African countries, Côte d'Ivoire and Ghana, produce over 70% of the world's cocoa beans [62]. Nevertheless, Africa consumes the least cocoa, accounting for about 4% of the global cocoa consumption. The largest consumers include Germany (45%) and the USA (27.5%) [3].

Several approaches to the application of blockchain technology to enhance transparency through data validation or access to real-time information in the cocoa sector have been reported in the literature (see Section 2.3) [58–60]. Nonetheless, existing approaches are not applicable to the world's largest cocoa-producing nations due to the limited infrastructure and low level of education of small-scale cocoa farmers. Moreover, the approaches pay little attention to the looming corruption and social issues that exist along the cocoa supply chain. Furthermore, the cocoa farmers, being the backbone of this sector, are mostly left out or poorly recognized in designing any solution to the issues facing the sector. This case study, however, takes into consideration all the above-mentioned issues by especially focusing on how to directly involve the farmers in any proposed solution.

### 3.1. Overview of the Cocoa Supply Chain

The cocoa supply chain is long and complex and involves different actors worldwide, such as cocoa farmers in Ghana and chocolate manufactures in Germany. Figure 1 gives an overview of the cocoa supply chain with its main actors, without considering certification programs such as UTZ or Fairtrade.

At the beginning of the cocoa supply chain, the farmer harvests the ripe cocoa pods and packs the cocoa beans into sacks, which are then bought by the Licensed Buying Companies (LBCs). At the LBC depot, the cocoa sacks are inspected and classified by the Quality Control Company (QCC). Subsequently, the LBC transports the sacks by truck to the Take-Over Center (TOC) of the Cocoa Marketing Company (CMC). Upon arrival at the TOC, another quality inspection takes place by the QCC, before the CMC temporarily stores the cocoa sacks at the TOC for shipment, during which period, the QCC performs a final quality check before the shipment is finally carried out. Subsequently, the cocoa sacks are exported by cargo ship (A) or transported within Ghana to processing companies (B).

In the case of option (A), the cocoa beans are exported to processing companies such as Barry Callebaut (Zurich, Switzerland) or Cargill (Wayzata, MN, USA). Major export countries are the Netherlands, Germany, and the USA, where the companies process the cocoa beans into semi-finished products such as cocoa liqueur, cocoa butter, or cocoa powder. Afterwards, the semi-finished products are transported to chocolate manufacturers such as Mondelēz (Chicago, IL, USA) , Mars (McLean, VA, USA), or Nestlé (Vevey, Switzerland), who process the semi-finished cocoa products into finished chocolate products. Then, the chocolate products are transported to retail stores worldwide, such as Aldi,

Tesco, or Carrefour. After all this, the end consumer buys the finished product in a retail store [11,63,64].

**Figure 1.** Cocoa supply chain.

In the case of option (B), the cocoa beans are processed into semi-finished products and partially into finished products in Ghana. After that, the semi-finished products are exported to chocolate manufacturers (like in (A)) or sold in local stores to the end consumers in Ghana.

### 3.2. The Cocoa Supply Chain Process in Ghana: From Beans to Export

The cocoa process from beans to export within Ghana involves three actors: the cocoa farmers, the LBCs, and the Ghana Cocoa Board (COCOBOD), including its subsidiaries. The COCOBOD is a government institution that regulates and controls the cocoa sector in Ghana. Beyond that, the COCOBOD grants licenses to private companies to purchase beans from individual farmers. These companies are called LBCs. However, only the subsidiary CMC of the COCOBOD is authorized to sell cocoa beans internationally [65,66].

The process from cocoa bean to export, which we describe in the following, was reconstructed from interviews conducted with a cocoa farmer, an officer of the COCOBOD, a district manager, and a research assistant working at an LBC. Based on these interviews, we analyzed the overall process (process flow, requirements, etc.) and built the proposed blockchain-based application.

The cocoa harvest season is from October to March (main season) and from May to August (mid-season). During the harvest season, cocoa farmers harvest the ripe pods every three to four weeks [67]. The cocoa process begins at the plantation with the ripe cocoa pods cut off, collected, and carried by foot to the pod-breaking point. After every single pod is broken and the good cocoa beans are selected, the farmer transports them to the fermentation area and, after the fermentation process, to her/his home. There, the cocoa beans will be dried on thinly stretched mats. Once the cocoa beans are dry, the farmer fills them into cocoa sacks and sells the full sacks to the LBC. Depending on whether the farmer lives in a rural or urban area, the responsible LBC employee (within an LBC, there are different positions, for which we did not differentiate due this being complicated) is called to pick up the cocoa sacks from the farmer or the farmer transports the sacks to the LBC depot by herself/himself, usually by taxi. The LBC employee weighs the sacks with her/his scale on-site to check the weight of each sack and to adjust it if necessary. Each

sack must contain 64 kg of cocoa beans. Then, the farmer receives a payment in cash or by mobile money and the LBC employee records the date, the number of sacks, and the amount paid to the farmer in both the farmers' cocoa passbook and in her/his own internal book. Afterwards, the LBC employee transports the cocoa sacks to the next LBC depot. The further process after the arrival at the LBC depot is described in Section 3.1.

### 3.3. Difficulties Faced by the Cocoa Farmers

Cocoa farming in Ghana is characterized by many modest farmers with small plantations. The COCOBOD estimates that about 800,000 families are involved in cocoa farming [68]. Farmers mostly live in villages and have their farms in the nearby areas. The average harvest of one farmer is between four and six sacks of cocoa per season.

Currently, there is no standardized system used by the COCOBOD or the LBCs for all cocoa farmers, with which the traceability of the cocoa sacks up to the farmer can be guaranteed. Therefore, it sometimes happens that the LBC buys cocoa beans from Côte d'Ivoire, which are of poorer quality and cheaper, and mixes them with the Ghanaian premium cocoa beans. The customer (located, for example, in Germany or Japan) typically notices this quality defect.

Based on the expert interviews conducted, it was determined that there is a high level of fraud when the cocoa sacks are sold to the LBC by the respective farmers. This includes the manipulation of the scales with which the LBC weighs the sacks or the declaration of the wrong number of cocoa sacks. Consequently, the farmer does not receive the cocoa price per sack to which she/he is entitled [5,69]. This is compounded by low and fluctuating cocoa prices [10]. The average annual income of cocoa farmers is between €560 and €800, depending on the size of the plantation or the number of cocoa sacks delivered [70].

Beyond that, the data, such as the number of cocoa sacks and their weight or the amount paid out, are entered only in handwriting in the respective notebooks and are not issued as a supplementary digital certificate. Furthermore, the QCC issues the certificates of the three quality inspections on paper and not digitally. This can not only lead to incorrect data or data shrinkage but can also be disadvantageous during evaluations.

Cocoa farmers in Ghana are generally males, over 50 years old, and have barely attended school. They live mainly in rural areas, which are characterized by poverty and poor infrastructure. This also affects their low acceptance of technical developments. There are hardly any farmers with internet-powered smart phones. Farmers typically have simple cell phones to make phone calls and send SMSs. The network coverage in the cocoa-growing areas is mainly provided by the two largest mobile phone providers, MTN and Vodafone. Nevertheless, there are places, especially in the rainforests and rural areas, where there is no stable network. A stable internet connection exists predominantly in the cities, whereas in rural areas, if at all, a weak internet connection is available [71].

### 3.4. A Blockchain-Based Solution for the Cocoa Supply Chain in Ghana

The characteristics of blockchain technology, especially the immutability of data and the already successful implementation in other agri-food supply chains, make this technology a good solution to promote sustainable transparency along the cocoa supply chain. The beginning of the cocoa supply chain is the most challenging, packed with issues, such as illegal cocoa plantations, human rights abuses (child labor, poor living and working conditions of farmers), as well as corruption and mistrust [9,11]. As mentioned above, the most pertinent issue is the situation where the farmers are cheated on by the LBCs in ways such as incorrect weights of cocoa sacks, leading to reduced payments made for their cocoa beans.

This paper presents a blockchain-based solution to make transactions between the cocoa farmers and the LBC more transparent and non-editable. In this context, transaction means the handover of the cocoa sacks from the farmer to the LBC, in return for payment. It is important that the cocoa farmers are integrated into the digital process using their own devices to avoid fraud. Therefore, their technical prerequisites need to be considered

(no technical devices, poor internet access, etc.), as well as the complexity of handling. Since cocoa farmers rarely use any technical devices with internet access, we developed an alternative way to document the transaction via SMS into a permissioned blockchain network, through which the data are saved immutably.

In the following, we describe the implementation of our blockchain-based solution for the cocoa supply chain. Overall, we made extensive use of open-source frameworks, where everyone has free access to the program code and which can be developed further or modified individually without any financial requirements.

Among the recognized open-source blockchain platforms, especially in the areas of supply chain and logistics, are Ethereum and Hyperledger Fabric [28,72]. For our application, we opted for Hyperledger Fabric (version 2.0) because this has unique characteristics, especially the permissioned blockchain within its private network. Thus, only authorized persons with the corresponding role profiles can access the blockchain network for the cocoa supply chain. In addition, Hyperledger Fabric is in use in the agri-food supply chains of, for example, Walmart, where it is used to map the supply chains of mangoes from the USA or pork from China [37,72]. A further reason is the modular infrastructure of Hyperledger Fabric, which is easy to expand and modify. The implementation was performed on Ubuntu 20.04.2.0 LTS, since this is open-source and it is easier to develop Hyperledger Fabric on it than, for instance, on the Windows operating system. Sending and receiving SMSs should be free for the farmers due to their poor living conditions. For this reason, an SMS-REST-API provider who offers two-way SMS with a Ghanaian mobile phone number would be advantageous. Currently, there are only three providers who can offer this service. Regarding long configuration times and high fixed and variable costs, we chose Twilio as the preferred provider. Twilio offers for the two-way SMS an international mobile phone number, which also works in Ghana. Besides Twilio's free test version, detailed documentation for different programming languages is available [73].

Before the farmers and LBCs can operate with the blockchain-based solution, they need to be registered in a database with information such as name, phone number, region, and role. After they have been registered, they are assigned automatically to an ID. Concerning the LBC employee, she/he additionally needs to create a password. In this case study, blockchain technology comes to its deployment when the cocoa farmer has cocoa sacks ready to sell it to the LBC. After the farmer has informed the LBC, an employee from the LBC comes to her/his home to pick up the cocoa sack. Now, the new process with the blockchain-based solution starts, as shown in Figure 2. First, the LBC employee has to login into the web app on her/his phone by entering her/his ID and individual password. Therefore, it is assumed that every LBC employee has a private phone, or a phone provided by her/his employer.

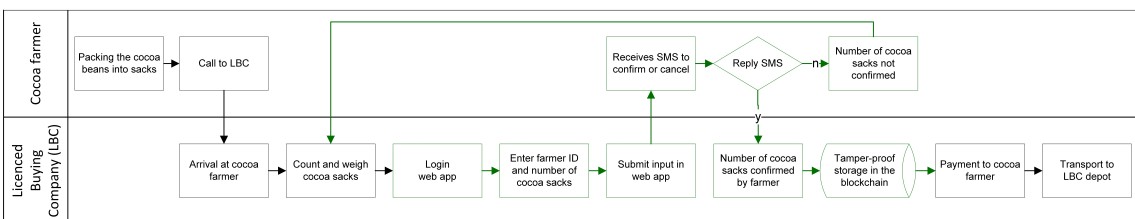

**Figure 2.** New process with the blockchain-based solution.

After logging in, the user will be directed to the main menu of the web app. From there, she/he can either create new transactions or inspect her/his previous transactions with a timestamp and statuses (pending, done, canceled, expired) in real-time. For generating a transaction, the LBC employee needs to enter the farmer's ID and select the correct number of cocoa sacks. When the LBC employee has filled in these two data, she/he can generate the transaction by pressing the *create* button. At the same time, the cocoa farmer receives an SMS with the transaction details, including the number of cocoa sacks (Appendix A). If the transaction details are correct, the user needs to confirm the transaction by replying to the

SMS with "y". Once a transaction between the LBC employee and the cocoa farmer has been successfully confirmed, the transaction data (names of LBC employee and farmer, number of cocoa sacks, payment, timestamp) are recorded in the blockchain. Simultaneously, the transaction is visible to all authorized actors that are part of the blockchain network. In case the LBC employee entered the wrong data, the farmer has the chance to cancel the transaction by sending an SMS with the content "n". Then, the transaction will not be saved in the blockchain to eliminate storage space. Consequently, the LBC can correct her/his input and create a new transaction. In case the farmer does not reply to the received SMS about a transaction within 15 minutes of receiving, the transaction expires and will not be saved in the blockchain. If a transaction has the status "done", "canceled", or "expired", both the farmer and the LBC employee receive an SMS about the status.

Figure 3 gives an overview of the blockchain-based solution's IT infrastructure. The blockchain-based solution was developed with open-source frameworks, except for a free test version of Twilio, as no budget was available for this case study. The LBC employee uses the frontend, i.e., the user interface of the web app, by logging in to access the homepage so that she/he can execute transactions or view her/his history there. The frontend is written in the HTML and Bootstrap languages with the use of Pug. The cocoa farmer uses her/his own cell phone, with which she/he sends an SMS. This SMS is routed to the backend server via a configured webhook from Twilio. For the backend server, Node.js was used, which interacts with the frontend, the SQLite3 database, and the SMS rest API/webhook of Twilio, but also with the blockchain network through an interface by its software development kid (SDK). The blockchain network stores confirmed transactions in addition to the blockchain in the CouchDB database. Successfully executed transactions can be accessed directly in the blockchain or via its world state, the CouchDB database. The blockchain-based solution runs on a local server and not on a web server yet, as this was sufficient for the case study.

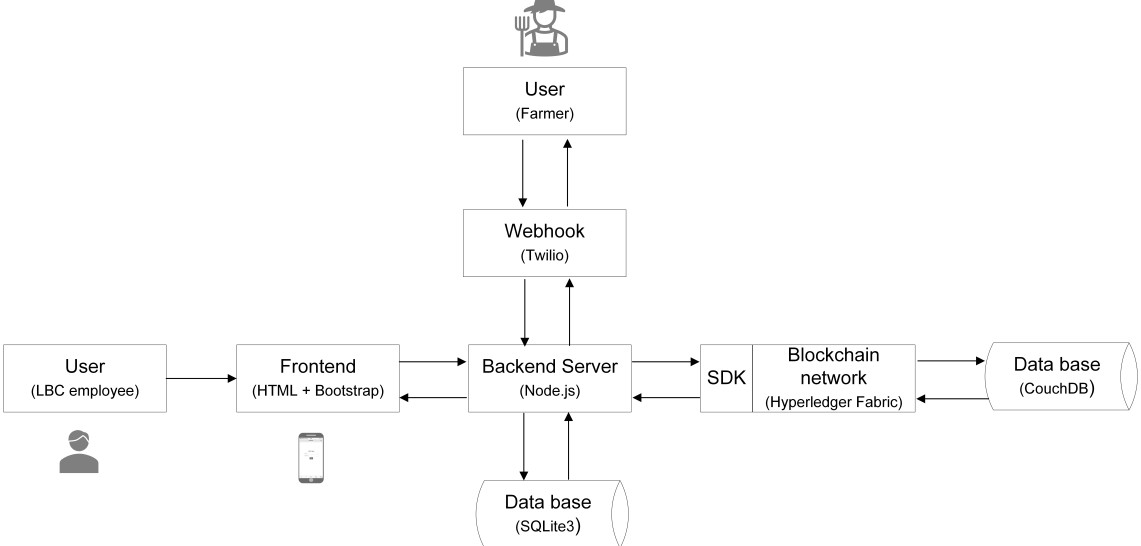

**Figure 3.** Infrastructure of the blockchain-based solution.

## 4. Discussion

In this paper, we demonstrate how blockchain technology can be used to support one step along the complex and long cocoa supply chain to establish a means by which fraud and corruption perpetrated against cocoa farmers can be tackled while promoting transparency. In addition, this paper demonstrates how high-end technologies, such as blockchain, can be implemented based on open-source frameworks without the need for significant financial investments.

Compared to the approaches in the literature, Arsyad et al. [58] used a two-factor blockchain that links legal documentation and blockchain technology within a traceability system via digital watermarking of the documentation media [58]. The downside of their

approach, however, is that it relies on textual and media-based documentation (photos, videos), which are rarely available or difficult to acquire due to the expensive and limited technologies in the developing world. Arsyad et al. [59] again proposed a farm transaction simulation implemented by a Modular Block Chain sensing instrument, which is coupled with equipment such as sensors and controllers [59]. This approach, though promising, would do little in fighting corrupt practices perpetuated against farmers in Ghana, Côte d'Ivoire, and other developing cocoa-producing nations. In addition, the proposed solution would be very cost-intensive considering the fact that the cocoa farmers in the focus country are small-scale farmers and the use of the technology devices mentioned above would be a financial burden to them. This solution can be applied in developed countries such as Japan but is less likely to be adapted in under-developed nations. Both approaches aim to attain transparency in the cocoa supply chain even though the solution proposed in this paper tackles corruption right from the endpoint of the farmers, taking into consideration the limited education and technologies available to the small-scale farmers in the focus region. Again, the approaches of Arsyad et al. [58,59] cover all aspects of cocoa farm management, whereas the case study in this paper focuses mostly on the farmer–LBC transaction after cocoa beans are harvested and bagged for sale.

Chong et al. [60] developed a blockchain system using a similar framework such as Hyperledger Fabric and programming languages such as Go and JavaScript, as used in this case study. The authors focused on the complete traceability of the products from the farmers until the product arrives in the warehouse of the importers [60]. Moreover, their solution also requires complex and expensive infrastructure that is not available to the farmers in the focus nations.

The main purpose of the case study in this paper was to develop a blockchain-based solution that makes use of cheap and readily available devices to fight corruption in the cocoa supply chain. Information about the origin of the farmer, cocoa sacks, weight, date and time can be safely stored in the blockchain and accessed by all players along the supply chain in real-time, similar to the approach by Chong et al. [60]. The model advocated in this paper can be extended to cover all aspects of the supply chain of cocoa beans from the LBC to the warehouse of the chocolate manufactures, like that suggested in the work of Chong et al. [60].

The presented application starts directly at the beginning of the cocoa supply chain considering the technical and logistical constraints of the cocoa farmers. At the same time, the beginning of the cocoa supply chain represents a critical point, which is characterized by gross corruption and fraud. The farmers generally own and use simple mobile phones, which mainly support phone calls and SMS only. Furthermore, cocoa farmers do not have access to the internet, which poses a challenge when using technologies such as blockchain, often in conjunction with the Internet of Things (IoT) or even simple mobile applications. In this respect, the intention was to develop the application in such a way that it can be operated by farmers without prior knowledge, experience, or internet connection. This is important to ensure that the data of the transactions are stored correctly in the blockchain, as both the LBC employee and the cocoa farmer must come to an agreement before a transaction is executed. The whole blockchain-based application was tested with a cocoa farmer in one of Ghana's main cocoa regions, Suhum in the Eastern Region. The case study showed that blockchain technology can be applied to the cocoa supply chain and that it is possible to use it with a simple cell phone to include the cocoa farmer in the blockchain process.

We assumed that each cocoa farmer can send an SMS with her/his mobile phone. Accordingly, we chose SMS as a means to send and receive data from and to any simple mobile device, which can be stored into the blockchain to foster easy practicability and adaptation and to include as many farmers as possible. The usability of new IT systems is one of the main impacts that leverages the user's intention and usage [74]. The farmers use their phones mainly for phone calls, but not to send an SMS. Therefore, they are most familiar with making phone calls and less with sending an SMS. For this reason,

an alternative to reply by SMS would be to communicate by voice call. This means that the farmer receives a phone call instead of an SMS, in which she/he is informed about the number of cocoa sacks. She/he could then confirm or cancel this verbally. However, it should be noted that due to the lack of school education, most cocoa farmers have little English knowledge. Therefore, this paper suggests investigating whether the existing English knowledge of cocoa farmers is enough to operate the application in a target-oriented manner, regardless of whether the communication takes place via SMS or voice call. It may also be possible to consider the adaptation of local Ghanaian languages both for voice and SMS responses instead of the English language.

Regarding fraud committed against farmers, the web app documents the number of cocoa sacks in the blockchain but does not consider the weight of the sacks. Since the weight of the sacks can be falsified by manipulated scales, it is recommended to develop a solution for this problem as well using the blockchain-based application to eliminate the risk of fraud. It is also important to mention that the blockchain-based application only shows how the number of cocoa sacks can be stored in the blockchain, but not how the cocoa sacks can be uniquely identified afterwards. One possibility would be to give the sacks a unique identification number, for instance as a QR code sewn onto the individual sack, which would also be documented during the transaction.

As a next step, besides the number of cocoa sacks, the money transferred between the LBC and the cocoa farmer could also be digitally mapped in the blockchain network. This can transparently ensure that the cocoa farmer receives a fair price for her/his cocoa beans. In Ghana, mobile money is a widely used means of payment. Therefore, there is a good opportunity to integrate this into the process so that, for instance, the transaction confirmation of the farmer can trigger the payment automatically. In this respect, it would be clear in the blockchain how much Ghanaian cedi (local currency) the cocoa farmer has received. Likewise, this can ensure that no fraud takes place during payment. Especially with regard to premium payments from certification programs such as Fairtrade or UTZ, this step could be very beneficial. Nevertheless, this requires an investigation into whether mobile money is also widespread among cocoa farmers. The status quo is that it is uncertain how willing the cocoa farmers and LBCs are to voluntarily use the blockchain-based application. Therefore, it is recommended to conduct extensive surveys with farmers and LBC employees to ascertain their readiness. Ultimately, it is essential for the cocoa farmer to receive fair payment for her/his sacks of cocoa. What happens later to her/his cocoa beans is of less interest to her/him. At this point, it is recommended to involve the COCOBOD and gain its acceptance to implement the blockchain-based application among farmers and LBCs and decide accordingly whether its use is voluntary or mandatory. This is also an opportunity for the COCOBOD to gain a better overview of the cocoa process within Ghana.

The COCOBOD, being a state-owned organization, currently regulates the production and sale of raw cocoa beans in Ghana. In addition to setting the selling price of cocoa each year, the COCOBOD is also responsible for developing programs that ensure the sustainability of the environment [65,66]. A widespread adaptation of the blockchain-based solution proposed in this work can only be possible through the acceptance and promotion by the COCOBOD. Nevertheless, its involvement could come with consequences, such as higher power in controlling and monopolizing the selling prices of cocoa beans, since it has already the export monopoly role with its huge political power [5].

It would also be advantageous for the COCOBOD to document quality inspections by its subsidiary QCC on the blockchain and to transfer the certificates digitally within the blockchain network. Anyway, for a large deployment of the blockchain-based solution in the cocoa sector in Ghana, it must be examined whether the open-source frameworks utilized in the case study can be implemented at a low cost on a large scale. As the complexity of the blockchain-based solution increases, due to the number of participants, more functions, as well as the development and improvement of the blockchain-based

solution may increase the costs, for example, through training, further development, and maintenance [55].

Blockchain technology for the cocoa supply chain has a great potential to add further functions, which were explained in the previous sections. In addition, there is a chance that the introduction of 5G networks by Vodafone and MTN in Ghana in 2023 will boost the blockchain-based solution through the possibility of better internet coverage, specifically in rural areas where farmers live, and in addition, a promise of unlimited and fast internet access and high data transfer speeds [75–78].

To tackle some of the challenges facing the cocoa sector, a blockchain-based solution was developed in this work, which mainly focused on the development of the process and technology for adaptability in the cocoa sector. Although a user response survey would make plain the adaptability of this solution, it is out of scope of this work and would be considered in the design of a further study.

In summary, the blockchain-based application enables the critical process steps at the beginning of the cocoa supply chain, where there is much fraud against the cocoa farmer, to be digitally documented and stored in a tamper-proof manner, thereby promoting transparency. Despite the opportunities for improvement of the blockchain-based application developed in this paper, it was shown that blockchain technology can be suitably implemented to promote transparency in the cocoa supply chain. This is because, during a transaction, relevant information, such as the number of cocoa sacks or timestamps, is made accessible to all stakeholders along the supply chain in real-time. The use of the blockchain-based application reduces fraud, as the farmer can now confirm the transaction, for example, the number of cocoa sacks, or cancel it in case of possible fraud before the payment is made. Furthermore, the developed application limits the illegal import and sale of lower-quality cocoa beans from neighboring regions, which are mixed with the farmer's cocoa beans to cover up the lack of quality. Since the blockchain-based application identifies each transaction and, thus, also the quality defect, it is possible to introduce restrictions on illegal imports and sales. Similarly, farmers working with certification programs, like UTZ or Fairtrade, can use the blockchain to document their certified sacks of cocoa. This information could also be made available to stakeholders such as chocolate manufacturers, for example, Lindt or Mondelēz, in the blockchain network.

The constraint of the results from this case study is that it focused only on one of the six cocoa regions in Ghana with a small number of farmers and LBCs, as compared to the studies published by Chong et al. [60] and Arsyad et al. [58,59], which covered a larger geographical area. Nevertheless, this case study highlights the possibility to implement blockchain technology in the entire cocoa supply chain of Ghana, by leveraging on its uniqueness of integrating all cocoa farmer in the region into the network while aiding the fight against corruption.

However, to gain transparency along the cocoa supply chain in the long run, it is necessary for all actors in the cocoa supply chain to be fully involved, which takes much time and cost to implement. For this reason, in the long term, the number of intermediaries should be reduced. In summary, this provides an opportunity to promote high transparency, minimize fraud, and subsequently, offer fair payments to cocoa farmers. Further research will be needed if all actors in the cocoa supply chain are willing to implement this solution.

## 5. Conclusions

Researchers have presented various approaches to make (agri-)food supply chains more sustainable, for example, refs. [41,60,79]. Blockchain technology has already been successfully implemented in other agriculture supply chains, such as coffee. This paper shows how to implement it in the cocoa supply chain to ensure transparency. We could integrate the farmers, who are otherwise neglected when it comes to the supply chain of cocoa, even though the supply chain begins with them. Faced with limited technology and other resources, we showed that it is possible to use simple SMSs coupled to the blockchain to promote transparency and cut down corruption at the same time.

The blockchain-based application that we presented in this paper documents all successfully executed transactions between the cocoa farmer and the LBCs, which are validated, non-manipulated, and saved into the blockchain. Due to the decentralized storage via the blockchain of transactions, as well as anti-counterfeiting, trust in the data of actors along the cocoa supply chain is improved. Authorized actors can also access the decentralized stored data in the blockchain at any time within the private blockchain network. It should be noted that this work identified critical issues relating to the blockchain-based application, for which recommendations for action were provided.

This solution constitutes a good basis to help the fight against corruption and bring about fairness in the cocoa sector. The presented solution for the farmers can be generalized and implemented in any other agriculture supply chain since it does not require internet connectivity or any extra skills from the farmers.

Finally, it should be mentioned that blockchain technology has great potential since it can be used not only for showing the number of cocoa sacks, but also in other process steps such as for money transfer or quality checks.

The unique methodology presented in this paper points out the need to pay more attention in the future to available infrastructure when designing solutions for specific geographical areas. Social issues can be better addressed by taking a more critical look at conditions on the ground and finding ways and means to develop an all-inclusive solution that integrates all actors along the supply chain in a specific food sector. This was well captured in this case study by concentrating on the incorporation of smallholder farmers into the supply chain of cocoa, without turning a blind eye to their inadequate educational background and technological infrastructure, to enhance transactional transparency between the farmers and the LBCs. In addition to this, the presented real case study enhances the field and value of blockchain technology for other agri-food industries, especially in developing countries with small-scale farmers.

Most of the challenges in agri-food supply chains, such as corruption, low payment, or child labor, especially in West Africa, affect the farmers' living and working conditions. Nonetheless, existing solutions to enhance transparency by implementing blockchain technology neither directly include smallholder farmers during their development and implementation, nor consider their limited infrastructural resources and knowledge [58–60]. Most of the raw products such as cashew or cocoa in West Africa are produced on a small scale and left out in generalized software solutions. The value of this work can be seen in its direct involvement of farmers and the simple approach to the challenges.

In the future, blockchain technology can promote transparency along the entire cocoa supply chain if all actors are included, even though it is challenging at the same time. Beyond that, for stakeholders, especially chocolate manufacturers in Europe, blockchain technology can be an interesting approach, especially considering the planned supply chain law in Germany, as well as in the European Union.

**Author Contributions:** Conceptualization, S.K.K.; methodology, S.K.K.; software, S.K.K.; validation, S.K.K.; writing—original draft preparation, S.K.K.; writing—review and editing, S.K.K.; visualization, S.K.K.; supervision, F.K.; project administration, S.K.K. All authors have read and agreed to the published version of the manuscript.

**Funding:** This research received no external funding.

**Acknowledgments:** Many thanks to Ernest Ahiavi, Karolis Valiusis, and all interview partners for your motivation and support.

**Conflicts of Interest:** The authors declare no conflict of interest.

## Abbreviations

The following abbreviations are used in this manuscript:

| | |
|---|---|
| CMC | Cocoa Marketing Company |
| COCOBOD | Ghana Cocoa Board |
| DTL | Distributed Ledger |
| IoT | Internet of Things |
| LBC | Licensed Buying Company |
| QCC | Quality Control Company |
| RFID | Radio Frequency Identification |
| SDK | Software Development Kid |
| TOC | Take-Over Center |

## Appendix A

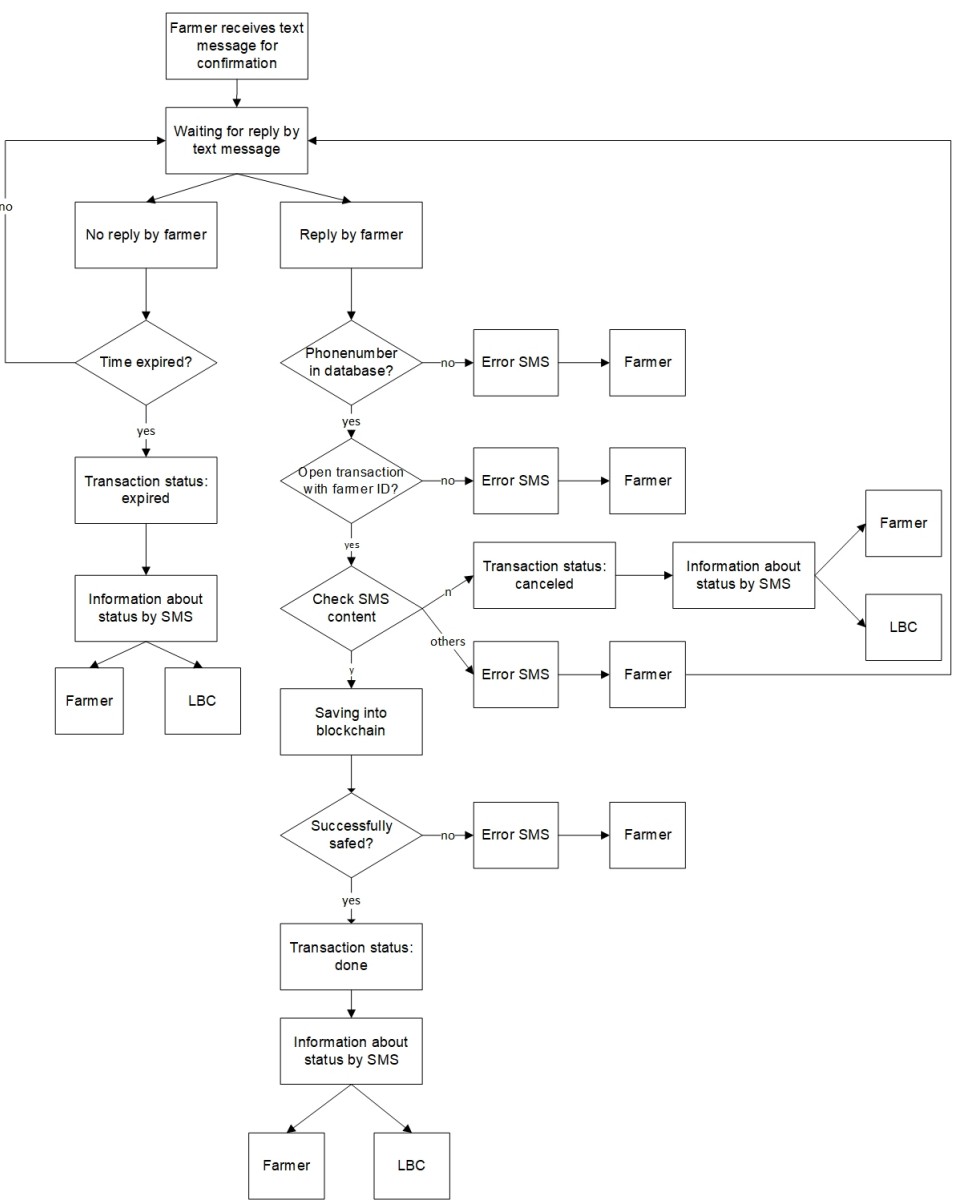

**Figure A1.** Detailed SMS process.

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
