# Peer review of "Can Blockchain Be a Basis to Ensure Transparency in an Agricultural Supply Chain?"

_sustainability, doi:10.3390/su14138044_

Round 1

Reviewer 1 Report

The authors have addressed an important and relevant research problem with a real-world case study. Following comments may be used to further improve the manuscript.

1. The scalability of the proposed approach may be discussed, particularly in terms of the cost and the computational complexity of using blockchain in a larger scale.

2. Some quantitative results to measure the effectiveness of the case-study can be included. For instance, a survey of the responses of the users can be analyzed to analyze the effectiveness of a blockchain based solution.

3. What are the techniques/technologies that may be used to improve the accessibility of the proposed solution to a broader group of people? The authors could discuss that aspect.

4. A high-level architectural diagram of the system used could be included. 

5. Could a technology driven solution may produce an unintended consequence of monopolies forming and have a negative effect in the long-run? The authors could discuss this issue. 

Reviewer 2 Report

I am pleased to have the opportunity to review this research paper. Although the topic of this research study is interesting and fits within the journal scope, I think authors should apply the comments indicated below to increase the quality of research justification, contributions, and findings. The manuscript knows lacks in scientific style and structure.

What is the originality of this research?  Paper research gap and originality should be better presented at the end of introduction section

Methodology

Pleas linked the methodology to the existing literature.

Discussion of results

What is different and what is common with the literature? Pleas connect the results to the existing literature and mention your contributions and what is new. 

Further Analysis

What is different and what is common with the literature? Pleas connect the results to the existing literature and mention your contributions and what is new. 

Conclusion:

-Managerial Implication

-Practical/Social Implications

Questions to be answered: What practical/professional and academic consequences will this study have for the future of scientific literature (theoretical contributions)?

Why is this study necessary? should make clear arguments to explain what the originality and value of the proposed model is. This should be stated in the final paragraphs of introduction and conclusion sections.

Additional references that can help you to improve the paper:

Bux, C., et al. (2022). Halal Food Sustainability between Certification and Blockchain: A Review. Sustainability14(4), 2152.

Akram, U et al. (2021). Impact of digitalization on customers’ well-being in the pandemic period: Challenges and opportunities for the retail industry. International Journal of Environmental Research and Public Health18(14), 7533.

Dasaklis, T. K., et al. (2022). A Systematic Literature Review of Blockchain-Enabled Supply Chain Traceability Implementations. Sustainability14(4), 2439.

Ionescu, C. A et al.. (2021). The New Era of Business Digitization through the Implementation of 5G Technology in Romania. Sustainability13(23), 13401.

Good luck!

Round 2

Reviewer 1 Report

The authors have sufficiently addressed the comments.

Reviewer 2 Report

Good luck!